# Disposal Practices of Unused and Leftover Medicines in the Households of Dhaka Metropolis

**DOI:** 10.3390/pharmacy9020103

**Published:** 2021-05-20

**Authors:** Mst. Marium Begum, Sanzana Fareen Rivu, Md. Mahmud Al Hasan, Tasnova Tasnim Nova, Md. Motiar Rahman, Md. Abdul Alim, Md. Sahab Uddin, Azharul Islam, Nuzhat Tabassum, Md. Marufur Rahman Moni, Rehnuma Roselin, Munny Das, Rayhana Begum, Md. Sohanur Rahman

**Affiliations:** 1Department of Pharmacy, East West University, Dhaka 1212, Bangladesh; rivu266@gmail.com (S.F.R.); tasnovatasnimhoque@gmail.com (T.T.N.); nurnahar1995@gmail.com (N.); nuzhatewu1@gmail.com (N.T.); marufurrahmanmoni@gmail.com (M.M.R.M.); rehnumaroselin@gmail.com (R.R.); 2Department of Public Health, North South University, Bashundhara, Dhaka 1229, Bangladesh; mahmud.hasan06@northsouth.edu; 3Institute of Synthetic Biology, Shenzhen Institute of Advanced Technology (SIAT), Chinese Academy of Sciences (CAS), Shenzhen 518055, China; motiar.rahman28@gmail.com; 4Department of Chemistry, Bangabandhu Sheikh Mujibur Rahman Science and Technology University, Gopalganj 8100, Bangladesh; alimbsmrstu@gmail.com; 5Graduate School of Innovative Life Science, Faculty of Engineering, University of Toyama, Gofuku, Toyama 3190, Japan; 6Department of Pharmacy, Southeast University, Dhaka 1213, Bangladesh; msu-neuropharma@hotmail.com; 7Pharmakon Neuroscience Research Network, Dhaka 1207, Bangladesh; 8Department of Pharmacy, Dhaka International University, Dhaka 1213, Bangladesh; islamazharul31@yahoo.com; 9Department of Pharmacy, Atish Dipankar University of Science and Technology, Dhaka 1212, Bangladesh; munnydas1987@gmail.com; 10Department of Pharmacy, Primeasia University, Dhaka 1212, Bangladesh; rayhana_kushum78@yahoo.com; 11Department of Biochemistry and Molecular Biology, Trust University, Barishal, Ruiya, Nobogram Road, Barishal 8200, Bangladesh

**Keywords:** leftover and expired medicines, households, medicine disposal pattern

## Abstract

**Background:** This fact-finding study aimed to attain an overall idea and knowledge about medicine disposal practices in Dhaka Metropolitan households. **Methods:** This mixed study (both quantitative and qualitative) was orchestrated to inspect the household leftover medicine disposal pattern’s governing status. A cross-sectional survey was conducted following a structured questionnaire and key informant interview with a household person and in-depth interviews with the top pharmaceutical and government officials. **Results:** Findings disclose that, for most of the key informants, the terms “drug disposal” and “drug pollution” were unknown; more precisely, 67% and 74% of key informants even did not hear these two terms. Almost all (87%) households faced undesired incidents due to the insecure storage of medicines. People disposed of excess and expired medication in regular dustbins (47%), threw out of the window (19%), flushed within commode (4%), burnt in fire (2%), and reused (4%). A good percentage of people (21%) returned unexpired drugs to the pharmacy and bought other medicines on a need basis. A total of 72% wanted a medicine take-back program, and 100% agreed on mass education on this issue. Officials of pharmaceuticals conferred mixed opinion: top-ranked pharmaceuticals will adopt leftover medicine disposal practices; middle and low-ranked pharmaceutical companies are reluctant, merely denied mentioning the less important issue. **Conclusions:** The absence of mass awareness and standard laws and policies may explain these existing aberrant practices.

## 1. Introduction

Over the past few decades, progressive advancement in pharmaceuticals has led to increased medication use [1,2]. The hike in usage caused significant issues revolving around accidental poisoning from unfitting use and pharmaceutical loading [3]. Despite being an important matter of concern, the disposal of unused medicines has long been an unspoken issue worldwide [4,5]. About 30 years ago, pharmaceuticals consisting of organic compounds that disturb human health and marine ecosystems were identified in the environment [4,6,7,8,9,10,11]. To date, a large number of studies showed the presence of pharmaceuticals in water bodies such as in groundwater [12,13], surface water [8,12], aquatic and coastline environments [13], and recipient water, sludge, and wastewater [14]. Most of the research in this area was performed in North America, China, and Europe [15]. Gauged concentrations of pharmaceuticals in the influents and seepage of wastewater treatment plants have revealed that these systems insufficiently remove some hazardous pharmaceuticals. [12,16,17]. Other studies [18,19,20,21] have shown pharmaceutical active ingredients and metabolites at nanogram levels in several potable water plants [22]. A Serbian report shows household antibiotics are the potential source of health and environmental risk [23]. Apart from being toxic to aquatic species, trace concentrations of pharmaceuticals have previously been spotted in drinking water in the USA and Greece [24] and cooked seafood [25,26,27,28,29]. Pharmaceutical waste includes hazardous and non-hazardous waste, controlled substances, and expired pharmaceuticals [30]. These wastes can come from manifold: from the manufacturing factories [31], from healthcare-generated waste, medical centers, and laboratories [32], and from unused household medications through evacuation after ingestion, injection, or infusion; washing out of topical medications during bathing; and discarding of unwanted or leftover pharmaceuticals [33]. The medical authorities discourage stocking drugs with an ill motive [34]. Storing unused drugs conceives a likelihood that someone will be exposed to them inadvertently and suffer from maltreatment [35]. In the United States, according to Poison Control Centers, approximately 23,783 out of 255,732 cases concerning inappropriate medication ingestion were associated with indecent exposure [36], where 5000 of them involved children less than six years of age [36]. Organizations and households might dispose of large amounts of drugs by putting them in landfills, returning them to the manufacturer, dumping the drugs into sewers, or using incineration [37]. Landfilling of active pharmaceuticals ingredients (API) reduces surface water release [38]. USA has adopted multiple methods to take back unused medicine under the National Association of Boards of Pharmacy (NABP) [39], from pharmacies, medical offices, dental offices, hospitals, manufacturers or distributors, and consumers [40,41,42,43]. In the UK, the new NHS Community Pharmacy Contractual Framework was developed in 2005. As per the National Health Service (Pharmaceutical and Local Pharmaceutical Services) Regulations 2013, local medicine stores are obligated to take back unused and excess medications generated in the households [44,45]. Canada has no precise nationwide drug take-back or disposal policy, but most provinces have their plans [46]. Most of the pharmacies in Canada take back unused and expired medications throughout the year [47]. The European Parliament released a directive vis-a-vis the disposal practices of medicinal products in 2001 [48]. In 2004, the EU circulated another directive that clarified the previous one and called for establishing medicine collection protocols [49].

Practices and awareness for proper disposal of leftover household medicine in the developed and developing middle-income countries and least developed countries are very disappointing and represented the same outcome of irresponsible disposal. The study from Yogyakarta province, Indonesia, Saudi Arabia, Vienna (Austria), Kumasi (Ghana), Israel, Plateau State (Nigeria), Tamil Nadu, and West Bengal, India revealed gaps and negligence existing in household medicine disposal, and creating awareness is needed equally for all countries regarding their financial status [50,51,52,53,54,55,56,57].

Proper disposing of medication is imperative for the safety of the household and the environment. To children, capsules and tablets mimic candy, often colorful and just the right size for popping into their small mouths. Further, teens, who are the ultimate experimenters, too often gain access to adults’ medication for pain, anxiety, or attention deficit disorder, with dire consequences.

Household drug disposal is a crucial matter to eliminate accidental exposure and maintain a safe environmental point of view. Bangladesh is a South Asian developing country that is inhabited by 163 million people, according to a World Bank report in 2019. This country’s pharmaceutical company is considered a multi-billion-dollar industry with a compound annual growth rate of 15.6% for the last five years, and expected growth is 15% for the next five years [58]. The “Research and Market,” a Dublin-based market analysis farm, reported that Bangladesh’s pharmaceutical market would surpass $6 billion by 2025. This report also focused on the healthy figure of Bangladesh pharma market, with >90% local company share, >75% generics drug share, and >$425 million export opportunity by 2025 [59]. In Europe, medicinal products’ consumption ranges from 50 g to 150 g/capita/year [60]. Based on this hypothesis, for Bangladesh, with a dense population and a number of world-class pharmaceuticals and availability of all kinds of medicine at a cheap rate, yearly medicine consumption must be high compared to any other country. Additionally, factors associated with human, biota, and environmental hazards, like sedentary medication purchase, shared medication by patients, non-adherence to medication regimens, and inappropriate use of drugs, lead to a significant amount of leftover household medicine and improper disposal. This improper disposal of medicine ultimately affects the overall ecosystem and water life with unimaginably negative consequences.

In Bangladesh, no previous study explored current disposal patterns and awareness of household medicines except one study investigating management medicine at hospitals [50]. This investigational study tried to generate preliminary evidence on leftover and household medicine disposal patterns and knowledge and practices at the household level. It investigated the role of pharmaceutical companies and associated government bodies in this issue. The study looked at the regulatory status of leftover household medicine disposal practices in 2019 throughout Dhaka Metropolitan by using mixed (quantitative and qualitative) research methods. Findings from this study are anticipated to assist in creating awareness about appropriate practices in households and trigger interest and attention among policymakers about formulating relevant regulations.

## 2. Materials and Methods

The data were obtained by triangulation of both quantitative and qualitative findings. The participants responded positively to the queries regarding their medication profile, e.g., either they follow the prescription, complete the full course, check the expiry date, follow the instructions and storage conditions, keep medicines away from children, read instructions on the leaflet, etc., like most other interview methods where respondents claim to abide by the ideal procedure. Quantitative data is organized into three different segments: current drug utilization practices, inappropriate practice for medication ingestion, and drug disposal knowledge and practice.

The quantitative and qualitative data generated from all procedures involved in this study are encapsulated below. 

### 2.1. Study Design

This study included three parts: a quantitative part (180 participants), a key informant interview part (KII, 180 participants), and an in-depth interview part (40 participants). The quantitative part was designed to obtain findings using structured questions; key informant interviews included a short interview of the same participants (180) who answered the questionnaire. An in-depth interview included interviews of 40 officials from both pharmaceutical companies and government bodies.

This study was conducted as a part of the academic research of the Department of Pharmacy, East-West University, Dhaka, Bangladesh. The relevant information sought was non-sensitive in nature, and a mixed-method (both quantitative and qualitative) was applied; hence a convenience sampling strategy was undertaken. A questionnaire containing twenty-one structured close-ended questions and one multiple-choice question was developed in this cross-sectional pilot study. Households were selected conveniently regardless of their socioeconomic status, gender, residence location. Structured questions reflected quantitative analysis and outcomes, whereas multiple-choice questions reflected qualitative findings. A brief key-informant interview (KII) was conducted with the participating household dwellers after the survey data collection was completed to get further insight. All participants in the quantitative part were requested to participate in KII, and participation was 100 percent.

Qualitative data were collected through in-depth interviews with senior government and pharmaceutical company officials. Two research associates from the 8th semester of the Department of Pharmacy’s undergraduate program, East-West University, assisted in this study as part of their undergraduate degree fulfillment. They first built a rapport with the participants to apply a dialectic approach to conduct in-depth interviews following the study protocol similar to Manzurul et al. [61].

### 2.2. Data Collection

The principal investigator trained two research associates for two months with a pilot-testing data collection procedure. Then, the final field-level collection of data took place for about ten months (from January 2019 to October 2019) in the areas nearby Rampura, Banasree, Khilgaon, Malibagh, Dhanmondi (Dhaka South City Corporation), and Badda, Banani, Mohakhali, and parts of Gulshan (Dhaka North City Corporation). Initially, data were collected (both qualitative and quantitative) from 300 households, but due to ambiguity and incomplete information, 120 sets were discarded, and the remaining 180 data sets were retained in this study.

### 2.3. In-Depth Interviews

A total of 20 senior officials, including policy-making officials, took part in an in-depth interview to gather information on multiple topics, like the idea about household leftover medicine, its disposal pattern, querying about adapting policies as a part of corporate social responsibilities, investment, and budgeting regarding household medicine take-back programs, etc. High-, middle-, and low-tier pharmaceutical companies were selected. Among twenty officials, five were director-level or policy-making personnel, and the remaining fifteen includes marketing strategies personnel, product development personnel, and others.

The second set of participants included 20 senior officials; among them, five were policy-making director level/equivalent ranked, five doctors, three professors, and others from Directorate General of Drug Administration (DGDA), Ministry of Health and Family Welfare, Government of the People’s Republic of Bangladesh, National Institute of Preventive and Social Medicine, Institute of Public Health (IPH), Pharmacy Council of Bangladesh, (PCB, autonomous), Department of Pharmaceutical Technology, Faculty of Pharmacy, University of Dhaka (Table 1).

### 2.4. Data Analysis

The data accumulated by surveying were analyzed mainly with simple descriptive statistics and represented in percentages, and information obtained from key informant interviews was represented as qualitative findings.

Data accumulated from in-depth interviews of top-ranked officials were analyzed after amalgamating themes and sub-themes based on the hypothesis. Information was formulated by the triangulation of data (Figure 1).

### 2.5. Questionnaire Item Development, Samples, and Population

The previously published articles, resources, and observation of existing practices associated with consumption/misuse and adherence to medications were the basis for questionnaire item development. As a preliminary, fact-finding, non-probability research approach, convenient sampling was employed. All households of the Dhaka metropolis were considered as populations.

## 3. Results

Table 2 represented the demographic data of both respondents, such as age, sex, marital status, educational qualification, and financial status. From Table 2, it was observed that female participants are prevalent 77% in the quantitative questionnaire and key informant interview, but male participants are more (70%) in the in-depth interview.

According to data obtained from current drug utilization practices, it was revealed that 81% of patients followed instructions while taking medicine, and 85% followed the direction for measurable medicine. Only 62% completed the patient’s full course of medicine. Inappropriate practices of medication ingestion were also distinctly noted. Surprisingly, 57% borrow and share their prescribed medicine with others, 58% of patients stop taking medicine after feeling better, and 26% follow the same outdated and obsolete prescription for recurrence. Drug disposal knowledge is less, and 72% of respondents opined to commence the medicine take-back program, but only 34% of respondents agreed to pay for the medicine take-back program.

However, when they were asked about their idea of drug disposal and drug pollution, 67% and 74% responded negatively as they had never really heard of the terms. Particularly, 87% of households experienced accidental exposure or undesired incidents due to unsafe storage and medicines disposal. As many as 72% of respondents are willing to participate in medicine take-back programs, but 66% are unwilling to pay for it. A total of 72% of household dwellers think it is the government’s responsibility to dispose of medicine in a safe place, and they told both north and south city corporation waste management departments could work for it. Only 66% of people follow a strict dosage regimen while taking medicine. A huge percentage (58%) of people stop taking medicine after they feel better, and 46% of patients consume leftover antibiotics. A total of 100% of respondents think public education is required to create awareness regarding leftover medicine disposal and disposal patterns. They all expressed the same opinion on this subject matter (Table 3).

Irrational prescribing of medicines and polypharmacy add to the burden of disposable drugs. Hence, this study included questions related to administration, dosing, and maltreatment of medicines. It has been found that 32% of participants buy antibiotics, sedatives, and sex-stimulating agents without prescription. Around 37% of respondents have little idea about wrong medicine intake, and 21% have no idea about the consequences of wrong medication intake. Some (24%) respondents are afraid of death due to wrong medication intake (Table 4).

Among the collective practices of medicine disposal, 47% of informants dumped them in the dustbin without any further processing, 19% threw them out of the window, 4% even reused the medicine, 2% burnt them in a fire, 21% returned them to the pharmacy for refunding, 4% flushed them in the toilet. Almost 2% of respondents followed two or more of these aforementioned methods, and another 1% claimed to have no leftover medicines in their houses. Though 47% of the respondents dumped medicine in the dustbin, they did not have any proper procedure to dump it. They threw in household rubbish boxes with other common garbage and did not think about harmful consequences on the environment (Figure 2).

### 3.1. Feedback from Key Informant Interview

Among 180 household respondents, most of all respondents disclosed similar practices regarding leftover medicine disposal. Since Metropolitan Dhaka city households were the source of data collection, most of the respondents were female. The age range of all the participants was between 18 to 65 years.

Their unequivocal remarks were illustrated as, “truly, I have no idea about systematic and disciplinary leftover medicine disposal, and if I add more, in real practice, our family members are used to consuming leftover medicine during recurrence of the same ailment.”—one respondent, a government administrative job holder by profession.

Another housewife echoed the same fact “actually, a common practice in our family is, if the medicine has within an expiry date range, we reuse it and if the date has expired, we just dump it in the rubbish box along with our other household garbage.”—a housewife.

Not all respondents replied the same; an undergraduate medical student said “after being admitted to medical college, one of my professors told me about this topic in class once, fortunately. After that, I told my family member to return leftover medicine to the pharmacy in exchange for money, in most cases, a trivial amount of money. I think this is a good practice”.

All 180 respondents echoed the same response and confessed that they have no regularized idea about systemic leftover medicine disposal. Most respondents expect a good guideline and awareness advertisement on this issue:

“As you know, in Bangladesh now, we have a good number of world-class pharmaceutical companies; they need to take initiatives about leftover household medicine disposal patterns through advertisement on electronic, social, and print media to make people aware of the aspect of corporate social responsibilities”—an army person.

There is a notable lack of awareness and information among indigent people, and this issue is non-understandable to them. A female garments worker on her weekend said “I never dump any medicine. Usually, I buy an exact number of medicines that the doctor or dokandar (salesperson of retail medicine shop) give me and take accordingly. Medicine costs me a lot of money, so I never waste”.

Overdosing and accidental poisoning was also reported as a significant issue during key informant interviews. To illustrate, one said “yes, I have a horrible experience, I was cooking, and my only six-years-old son drank a whole bottle of syrup that a doctor prescribed during his dengue fever; after that, I never store any leftover medicine for reuse. I throw all medicine in the dustbin and sometimes through a window”.

### 3.2. Feedback from an In-Depth Interview of Top-Ranked Officials of Pharmaceutical Companies

High-, middle-, and low-tier pharmaceutical companies’ top management officials were interviewed in-depth according to their availability. Pharmaceutical companies with a good portfolio exhibited a positive attitude regarding leftover medicine disposal.

“We could adopt good policies and invest in leftover medicine disposal as well as the collection of leftover medicine by our workforce as a part of corporate social responsibilities. We are a big conglomerate and I do believe there are many things we can do as an exemplary event”—a CEO of the top-ranked pharmaceutical company. 

There is no government policy and direction for the pharmaceutical companies to work on household medicine disposal. To embellish “we until now get no direction or obligation from the part of the government to do work on household medicine disposal awareness or collection such that, if the government directs us, we definitely will do so accordingly”. A market strategist of a big pharmaceutical company.

Educationists opined on environmental issues and emphasized enacting the law as early as possible on household medicine disposal patterns. Echoed voice “disposal of any chemicals, either pharmaceutical waste materials or finished dosages are equally harmful to a balanced ecosystem. So, we need to adopt policies and initiatives by both government and pharmaceutical companies”—a professor from the University of Dhaka.

Low- and middle-tier pharmaceutical companies were reluctant to consider the household leftover medicine take-back program, considering the medicine take-back program ancillary, and investment is another issue.

“In Bangladesh, the pharma-market is very competitive, and nearly all companies invest more than 20% of their budget on promotional activities, so investment as a routine part in the leftover medicine take-back program will be an extra burden for us.”—a managing director of a middle-tier company.

Some officials judged the medicine take-back program as a less concerning issue. In words “There are other numerous issues like large-scale pharmaceutical waste management, maintaining effluent treatment plants, etc. which are more important issues rather than the household leftover medicine take-back program.”—a top-ranked official of a low-tier pharmaceutical company.

### 3.3. Feedback from the In-Depth Interview of Top-Ranked Officials of Government Officials

The top two monitoring institutes of the Bangladesh government, namely the Directorate General of Drug Administration (DGDA) and the Ministry of Health and Family Welfare, deal with pharmaceutical companies’ various concerns and routinely monitor and follow up their legislation-related activities. However, to date, there is no policy regarding the household leftover medicine take-back program.

Honestly, I admit, like first-world countries, we have no policy on the leftover medicine take-back issue by pharmaceutical companies. It is high time to adopt a law on this to reduce environmental pollution through small scale. A director of a government organization.

A public health hazard is another major issue of accidental medicine ingestion within households. A public health expert echoed this matter. According to him “just implementing a single rule could lessen the phenomena of accidental leftover medicine if the government wants. Pharmaceutical companies will collect leftover medicine once a month and will destroy those systematically. In that case, pharmaceutical companies will adopt that policy as a part of corporate social responsibilities.”—a public health expert and government doctor.

## 4. Discussion

Accidental poisoning by pharmaceuticals kept in households is a less-discussed issue in most Asian countries; Bangladesh is not an exception. People are becoming concerned about the safe use of medicine, but they are unaware of safe disposal in contrast. Our study’s findings show that knowledge and awareness about leftover household drug disposal in Dhaka, Bangladesh is alarmingly low. Following prescriptions properly is a prerequisite to following a dosage regimen and ensuring the rational use of drugs. Ensuring this will minimize the piling up of unused medications to a greater extent. This can only be achieved by raising consciousness on the following prescription correctly. Our studies found that only 94% of participants claimed to be taking medicine after being prescribed it by the doctor, and 32% of respondents buy antibiotics, sedatives, and sex stimulants without prescription, which may lead to the public health hazard [61].

The findings of our study regarding drug disposal practice generate distressing evidence. In Dhaka Metropolitan, almost 67% of respondents have no idea about drug disposal. Though the scenario was also similar in developed countries, even in the late 20th century, proper infrastructures and jurisdiction have significantly improved their situation in the last few years [62]. In 1996, a study found 98% of the USA respondents kept unused drugs in the household. However, following the Drug Enforcement Administration (DEA) establishment, a National Prescription Take-Back day is being celebrated to collect unused medications [62]. They instruct those who are unable to travel to reach them to mix medicines with inedible substances, place them in sealed containers, and dispose of the mixture [63]. Despite only one government-funded program for returning unwanted/expired medications to retail pharmacies (The Return Unwanted Medicines, RUM) since 1998, the systematic medicine take-back program is satisfactory in Australia and the USA [64,65]. In contrast, in Croatia, pharmaceutical waste disposal services are not sufficient [65]. In New Zealand, a third party is appointed by pharmacists to collect and destroy unused and expired drugs [66].

However, in developing and underdeveloped countries, the situation is even worse. In a study in Ethiopia, even though most participants (58%) are aware of the hazards resulting from improper drug disposal and prefer FDA and WHO disposal methods (75%), this does not comply with existing Ethiopian practice. About 21% keep unused medicine at home [67]. The study revealed that Ethiopia has 67% of respondents who do not know about drug take-back programs similar to this study [68]. “Drug Disposal Day” could be a practical approach for taking back medicine [69]. Another study in India revealed that 96% of consumers threw away expired medication, and 73% were discarded in household trash with other garbage [70], supported by another study of Australia and Ethiopia [71,72]. According to this study, 72% of respondents emphasized safe drug disposal similarity with a study of Malaysia for household pharmaceutical waste (HPW) [73] and a similar study reported in Poland and Lebanon [74,75]. Non-proper storage of medicine was reported in 30% of respondents in our study, supporting medicine’s home storage behavior, a previous study in China, USA, and [76] found. Medication waste management has become a major challenge to the healthcare sector in Bangladesh [77]. In this study, the general ways of drug disposal among the participants were also hazardous (Figure 2). The study in Ethiopia shows that throwing into the trash (16.4%), flushing into the toilet (13.3%), and throwing into the environment (10.4%) are the common practices of drug disposal [67], and in Turkey and Australia, household medicine was also disposed of inappropriately, of [78,79]; the same scenario showed in Bangladesh. Another study in Kabul, Afghanistan shows that even though 98% of the respondents are aware of improper drug disposal practice’s negative environmental consequences, around 77.7% of them discarded unused medicines directly in household garbage [80].

It is high time to enforce strict legislation to stop such malpractices. In developing countries like Bangladesh, the scenario is very distressing. A study among 143 families from urban and rural settings together in Serbia showed that the most common drug disposal method was throwing into the garbage (80.3%) even though half of them were aware of the harmful consequences of their exposure to the environment [81]. Another study in Saudi Arabia found that 28% of participants keep unwanted and unused drugs in the home, and only one-third of them used to return those to the pharmacy [78]. In our study, around 79.4% of participants kept the unused drugs out of reach of children in a household, much higher than that of Serbia (11%) [81].

Some feasible interventions like storing leftover medicines on a high shelf out of sight of children will help protect them, and regular disposing of unused and expired drugs will reduce risks for all ages. Mailing or returning leftover medicines to the pharmacy, using a disposal kiosk, carefully trashing them, and flushing out liquid medicines can be some disposal methods [35]. The manufacturers should use tamper-resistant boxes to allow the return of unused medicines in a pharmacy. Not only medicines, but also medical apparatuses, like syringes, saline bottles, ampules, vials, tubes, must be disposed of safely and immediately after use. It is challenging for governments to discuss how to dispose of expired drugs [82] properly. Pharmaceuticals are preferably disposed of at high temperatures (i.e., above 1200 °C) for incineration. Such facilities, equipped with adequate emanation control, are mainly found in the modern industrialized world. Expenditures for incinerating the pharmaceutical waste in Croatia and Bosnia and Herzegovina range from USD 2.2/kg to USD 4.1/kg [36]. Whereas, when respondents were asked about paying for the drug return activity run by the government, 66% were unwilling to allocate any budget for that; rather, they are interested in returning medicines to the pharmacy on their own, which they often fail to do before an accident happens.

In this study, 58% of respondents skip medicine after feeling better, which supports a study from Cyprus where consumers reduce or take off medicine after feeling better [83]. Before implementing any intervention, people should first be educated on the necessity of disposal of pharmaceuticals. All the participants agreed to create awareness among mass people, and 72% think the government should take the responsibility to do so. However, the local communities and councils, and most importantly, pharmaceutical companies, should play a crucial role. The best source of advice can be the community pharmacy and local government corporation [35]. Amidst the attempts taken in the United States to inspire consumers to bring their leftover drugs to disposal centers, instructing individuals and related organizations about the effective ways to dispose of drugs was one of the first lines of effort [84]. An in-depth interview with the top management of pharmaceutical companies ultimately reveals a mixed phenomenon. Low- and mid-tier companies think household medicine disposal practices are a less important issue, and high-tier will contribute positively as part of the social responsibilities of leftover household medicine management. Government organizations are completely unaware of leftover medicine disposal’s commencement, and they do not adopt any policy or awareness program among city dwellers.

Drug disposal poses a dilemma about optimizing the twofold protection of ecological health and human well-being [33]. This study also approaches several goals like good health and well-being, clean water and sanitation, responsible consumption and production, climate action, life below water, and life on the land of sustainable development goals (SDGs). The SDGs are considered a “blueprint to achieve a better and more sustainable future for all” to be carried through by 2030. The segmentalized preservation of natural phyto and fauna, biota, and human lives is a matter of close concern. So leftover household medicine disposal practices need considerable attention in Bangladesh to manage a safe life and environment. The pharmacies already manage drugs and can recommend what type of disposal method is the most appropriate. They can stockpile drugs and dispose of them as a community service. Some regions may have special government services that offer to take back and dispose of drugs. Sometimes, either a pharmacy or a government service will provide a way to dispatch drugs to a special drug disposal facility [35]. A study from the USA stated that 80% of respondents do not get any appropriate medicine disposal [85]. Among various approaches to public education and awareness, the establishment of legislative bodies and “Polluter Pay” could be some options [54]. If there are no other options and one must put drugs with other rubbish, a recommended securer way to do so is by mixing the drug with unattractive trash. Flushing pills into the toilet can trigger drug pollution, but this might be a justified way to rapidly dispose of hazardous drugs. Seepage from the primary package of medicine after swelling and degradation increases environmental contact but disposing to bins increases the chances of human or pet exposure, though inadvertently and via diversion. While pragmatic procedures for drug disposal need to be extensively promulgated, an optimum and sustainable solution would be diverse alterations in the healthcare system to minimalize the frequency of leftover drugs by subsiding the number of medicines prescribed and dispensed [33].

Several practical approaches could improve these malpractices. These are: creating awareness among mass people by involving all kinds of media (print, electric, and social), policy-making by the appropriate body for legislation and regulation, involving climate activist groups or institutions enabling safe leftover medicine disposal campaigning, inclusion of specific information and warnings regarding safe disposal of medicine in the label and secondary packages, encouraging of returning medicine to the pharmacies, and leftover medicine collection by crowdfunding, etc. 

## 5. Conclusions

Neither the government bodies nor the pharmaceutical companies are currently concerned about the safe and secure disposal of unused medicines in Bangladesh’s households. Until 2019, there was no protocol regarding leftover household medicine disposal. Medicines being thrown into the environment without following any safety protocol pose a major threat to the ecosystem. Prudent and immediate measures are required to get rid of this predicament, which can be initiated by educating mass people, by involving print, electric, and social media, and by policy-making about the onslaught of adverse events of unsafe drug disposal. Raising awareness among households regarding the negative consequences of accidental drug exposure and improper disposal will pave the way for the concerned authority to take initiatives to implement rules.

## Figures and Tables

**Figure 1 pharmacy-09-00103-f001:**
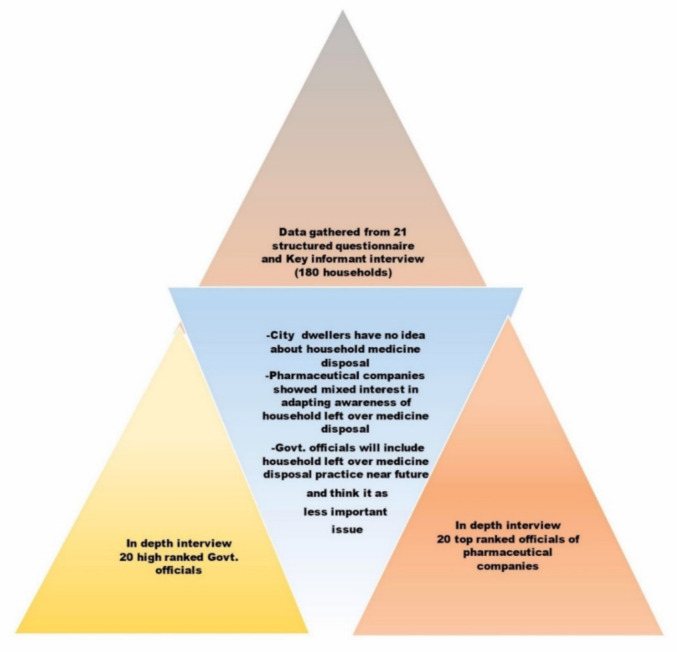
Data triangulation is related to core findings.

**Figure 2 pharmacy-09-00103-f002:**
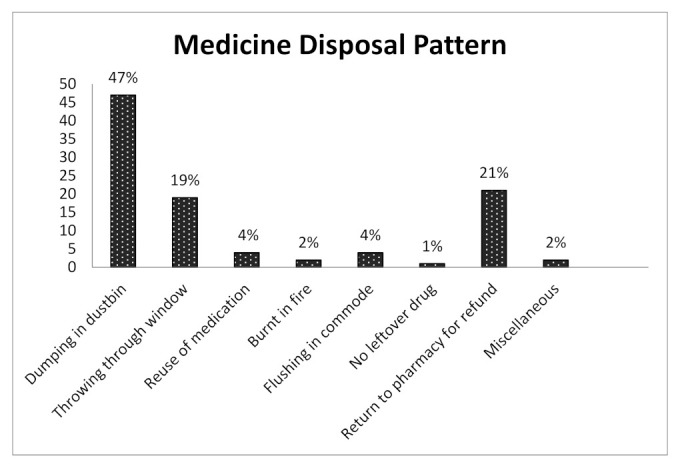
Common medicine disposal practices in the household.

**Table 1 pharmacy-09-00103-t001:** Categorized sources of respondents of the study.

Sources of Data Collection	Sample Size
Structured questionnaire and key informant interview	180 participants from households
In-depth Interview	40 top-ranked officials
a. Authorities	6
b. Policy-making personnel from pharmaceutical companies	7
c. Non-policy making top-ranked personnel from pharmaceutical companies	13
d. Policy-making personnel from govt. organization	5
e. Non-policy making high officials	7
f. Doctors (general practitioner)	5
g. Professors of pharmacy	3

**Table 2 pharmacy-09-00103-t002:** Demographic data of key informants/participants who responded in quantitative questionnaire and key informant interview (180).

Parameters	Key Informants/Participants Who Responded in Quantitative Questionnaire and Key Informant Interview (180)		Participants Who Responded in the In-Depth Interview (40)	
	Male	Female	Male	Female
Gender	42 (23%)	138 (77%)	28 (70%)	12 (30%)
Age range				
18–29	4 (10%)	23 (17%)	3 (11%)	2 (17%)
30–41	9 (21%)	67 (49%)	7 (25%)	4 (33%)
42–53	12 (29%)	26 (19%)	11 (39%)	5 (42%)
54–65	17 (40%)	22 (16%)	7 (25%)	1 (8%)
Educational qualification				
No general education	5 (12%)	17 (12%)	0 (0%)	0 (0%)
Primary education	12 (29%)	43 (31%)	0 (0%)	0 (0%)
Secondary education	19 (45%)	57 (41%)	0 (0%)	0 (0%)
Graduated	6 (14%)	21 (15%)	28 (100%)	12 (100%)
Marital status				
Married	22 (52%)	85 (62%)	25 (89%)	12 (100%)
Unmarried	17 (40%)	48 (35%)	2(7%)	0 (0%)
Others(widower/widow)	3(7%)	5 (4%)	1 (4%)	0 (0%)
Financial condition				
Poor	7 (17%)	18 (13%)	0 (0%)	0 (%)
Lower-middle class	11 (26%)	21 (15%)	0 (0%)	0 (0%)
Middle-middle class	15 (36%)	54 (39%)	0 (0%)	0 (%)
Higher-middle class	7 (17%)	33 (24%)	19 (68%)	8 (67%)
Rich	2 (5%)	12 (9%)	9 (32%)	4 (33%)

**Table 3 pharmacy-09-00103-t003:** Disposal and storage patterns of medicines in the households.

Information Obtained from Questionnaires	Percentage (Yes)
a. Current drug utilization practices	
The patient claimed to be taking medicine after being prescribed by the doctor	94
Completing the full course of medicine as per the doctor’s prescription	62
Following the instructions, while taking medicines	81
Keeping medicines out of reach of children	79
Reading the insertion/drug literature before taking medicines	56
Following dosage regimens, while taking medicines	66
Storing medicines according to the directions for proper storage conditions	70
Follow the direction of measurable medicine	85
b. The inappropriate practice of medication intake	
Borrowing or sharing own medicine with others	57
Stopping medicine intake once starting to feel better	58
Consuming the leftover antibiotics	46
Following the same prescription for taking medicine after facing a recurrence of symptoms	26
c. Drug disposal knowledge and practice	
Checking the expiry date of medicines before throwing them out	91
Throwing out the leftover medicines	33
Having any idea about drug disposal	33
Having any idea about drug pollution	26
Experiencing incidences regarding unsafe drug disposal in own household	87
Having desire to take part in any medicine take-back program	72
Having the willingness to pay for that	34
Public education is required to create awareness	100
It is the government’s responsibility to dispose of medicines in a safe place	72

**Table 4 pharmacy-09-00103-t004:** Issues related to medication administration, dosing, and mistreatment.

**Habits Regarding Buying Medicine Without a Prescription**						
	Buying Antibiotics, sedatives, sex stimulants (including OTC) without prescription			Buying all types of medicine (including OTC) with prescription		
Number	58 (32%)			122 (68%)		
**The Idea about Medication Intake**						
	No ideas about wrong medication intake	Having less idea about wrong medication intake	Have a proper idea about wrong medication intake		Being afraid of death due to wrong medication intake.	Miscellaneous
Number	38 (21%)	67 (37%)	25 (14%)		44 (24%)	6 (4%)

## Data Availability

Not applicable.

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
