# Peer review of "Disposal Practices of Unused and Leftover Medicines in the Households of Dhaka Metropolis"

_pharmacy, 2021, doi:10.3390/pharmacy9020103_

Round 1
Reviewer 1 Report
The authors addressed the reviewers' comments satisfactorily.
Author Response
Thank you.
Reviewer 2 Report
In my opinion, your article is of interest for the journal, however some major changes are still required in order to improve the scientific soundness of the paper.
Introduction:
I suggest to restructure the introduction as follows:
- Brief introduction of the problem worldwide (contamination, regulation, and management), with the specification in developing middle-income countries
- How is the problem in Bangladesh and what are the main issues detectable
- Introduce why your study could support in solving the issues listed in Bangladesh, specifying the objective and the method used. Why exactly this method? Why it is useful for your investigation? What is main impacts that you want to obtain after the data gathered
Methods
I suggest adding a subsection within the method section where you describe items developments and samples.
Tables with items of questionnaire must be presented. How did you choose the items? The draft of the questionnaire was pilot-tested?
Can you specify how you chose the households? Why did you initially choose 300 households? Is their number representative for the studied area? Is the final sample representative for the population of Dhaka Metropolitan? How was it calculated? What was the participation rate in the study? 100%?
Table I in section 2.1. should be moved to the results section
Results
The results should be presented in a more concise manner. The results presented in the tables must not be repeated in the text; they are interpreted in the discussion section
The text from lines 169-170 and 176-183 should be moved to the methods section
Discussion
You should better explain the context where the analysis took place, why your study is new in comparison with the previous studies and why it provides useful information.
Reviewer 3 Report
The authors did an intensive review on the causes of unused and leftover medicines and its relationship to drug utilization. Drug disposal is a public and global health issue. The United Nations approved Sustainable Development Goals (SDGs) in 2015, which are a collection of 17 interlinked global goals for us to achieve a sustainable future life. The topic of unused medication disposal is relevant to many of the goals including lives under the water, better health, economic development, etc. It is a timely topic both in developed and developing countries. The authors should include SDGs in the discussion to bring the attention of this issue to an international audience. Please see my comments below: 1. This is essentially a descriptive study, which surveyed and interviewed convenient citizen as well as selected government and industry representatives. Please explain what the exact “convenient” sampling scheme is. 2. I do not dispute most of the findings. The authors should have explored the relationship between demographic background and drug utilization practice, and drug disposal knowledge and practice. Can you combine quantitative (survey) and qualitative results (interview) to come up with an in-depth insight of the issue? 3. The authors should propose a system to improve drug disposal with the understanding of today’s challenging health care environment and consider topics such as open systems for medication purchases, shared medication by patients, non-adherence to medication regimens, and inappropriate use of medication. 4. The authors should investigate and report those households (120 out of 300) who refused to participate in the survey the interview. 5. In figure 1, the authors should have included twenty-one item questionnaire survey in the top small triangle describing key informants. 6. The authors should state the reasonable expectations of medication disposal outcomes considering the challenges of a socioeconomic environment and less regulated pharmaceutical utilization environment in a developing country. 7. The authors should explain the reasons why there is a lack of knowledge of medicine take back programs and proposed a feasible policy for Bangladesh. Discussion should consider that the survey and interviews were conducted in a much well educated and affluent segments of the country. 8. Discuss the resources to support a feasible drug disposal initiative in a developing country. The country should understand the priority and balance conflicting goals in the system. Patient safety, awareness of environment, economic environment, SDGs etc. should all be considered in drafting a sound policy for consideration. 9. The indifference to support the medication disposal program by middle to small size pharmaceutical companies is understandable due to the limited resources. It is difficulty to expect these companies to deploy 20% of revenues for promotion of medical disposal programs in a competitive low margin market while asking them to deal with an invisible public good such as education public for drug disposal. The public policy should kick in to resolve this issue if all parties can be in the same level of playing field. You may apply surcharge for every pill sod if the health care system can afford it. The authors should also explore what big or top ranked pharmaceutical companies can do and have done based on the statement in lines 267-270. "We could adopt good policies and invest in leftover medicine disposal as well as the 267 collection of leftover medicine by our workforce as a part of corporate social responsibili-268 ties. We are a big conglomerate and, I do believe many things we can do as an exemplary 269 event"- A CEO of the top-ranked pharmaceutical company."Author Response
Please see the attachement.

Round 2
Reviewer 2 Report
The authors have reviewed all the aspects reported and the manuscript has been significantly improved.
This manuscript is a resubmission of an earlier submission. The following is a list of the peer review reports and author responses from that submission.
Round 1
Reviewer 1 Report
This paper presents the results of a survey about the management of unused and leftover medicines in Dhaka.
The authors write a good literature review, however, in my opinion, some weaknesses of the manuscript have to be addressed.
- Your sample is a short focus group. Is this adequate for safe and secure conclusions?
- No information about the socio-economic profile of the sample is provided. You could run some statistical test to empower your findings.
- No suggestions, implications consistent with the results are provided. How can the practitioners, the policymakers in the field use your findings? What are the limitations of your research?
Reviewer 2 Report
General aspects
First of all, it is unusual as 15 authors to write a paper with less than 10 pages.
The content is not presenting any new/special aspects, just a particular case regarding medicines waste management/practices. Moreover, the language used requires a serious revision in terms of expression/repetition/terms.
Particular aspectes
The sample size is also not very relevant, and the jobs of the interviewees don't really matter.
Table 2. It is obvious that in the second column, if percentages are presented (as it is mentioned in the table header), there was no need to mention the % symbol next to each of the values inserted in that column.
Regarding the questionnaires, the following aspects are not presented:
- What is the objective of your study / main impacts that authors want to obtain after the data gathered.
- How did the authors choose the items? Did the authors classified the items? Which list did you consider? You should better motivate each item
- How were the criteria for including the interviewees established?
- Who validated the questionnaires?
- If the sample that completed the questionnaires is statistically representative, how was this fact established?
- Was the questionnaire pretested?
There are many references for such a short work. However, about half of them are older than 10 years, and the literature abounds with recent papers on the topic (including in MDPI journals) - some examples bellow:
https://doi.org/10.1016/j.jenvman.2015.08.013
http://dx.doi.org/10.18203/2319-2003.ijbcp20164081
https://doi.org/10.3390/su10082788 https://doi.org/10.1016/j.wasman.2015.12.020 https://doi.org/10.1186/s12889-016-3975-z https://doi.org/10.1289/ehp.8315 Tit et al., Disposal of unused medicines resulting from home treatment in Romania, J. Environ. Prot. Ecol., 17(4), 2016 From my point of view, the work requires an extensive restoration/reshape/completions/ deepening especially the methodology regarding everything that means questionnaires and the presentation of the results.